# Dope Dyeing of Regenerated Cellulose Fibres with Leucoindigo as Base for Circularity of Denim

**DOI:** 10.3390/polym14235280

**Published:** 2022-12-02

**Authors:** Avinash P. Manian, Sophia Müller, Doris E. Braun, Tung Pham, Thomas Bechtold

**Affiliations:** 1Research Institute of Textile Chemistry and Textile Physics, University of Innsbruck, Hoechsterstrasse 73, 6850 Dornbirn, Austria; 2Department of Pharmacy, Pharmaceutical Technology Section, University of Innsbruck, Innrain 52/c, 6020 Innsbruck, Austria

**Keywords:** indigo, cellulose, recycling, circularity, viscose fibre, denim

## Abstract

Circularity of cellulose-based pre- and post-consumer wastes requires an integrated approach which has to consider the characteristics of the fibre polymer and the presence of dyes and additives from textile chemical processing as well. Fibre-to-fibre recycling is a condition to avoid downcycling of recycled material. For cellulose fibres regeneration via production of regenerated cellulose fibres is the most promising approach. Textile wastes contain dyes and additives, thus a recycling technique has to be robust enough to process such material. In an ideal case the reuse of colorants can be achieved as well. At present nearly 80% of the regenerated cellulose fibre production utilises the viscose process, therefore this technique was chosen to investigate the recycling of dyed material including the reuse of the colorant. In this work, for the first time, a compilation of all required process steps to a complete circular concept is presented and discussed as a model. Indigo-dyed viscose fibres were used as a model to study cellulose recycling via production of regenerated cellulose fibres to avoid downcycling. Indigo was found compatible to the alkalisation and xanthogenation steps in the viscose process and blue coloured cellulose regenerates were recovered from indigo-dyed cellulose. A supplemental addition of reduced indigo to the cellulose solution was also found feasible to adjust colour depth in the regenerated cellulose to the level required for use as warp material in denim production. By combination of fibre recycling and indigo dyeing the conventional yarn dyeing in denim production can be omitted. Model calculations for the savings in water and chemical consumption demonstrate the potential of the process. The proportion of the substitution will depend on the collection rate of denim wastes and on the efficiency of the fibre regeneration process. Estimates indicate that a substitution of more than 70% of the cotton fibres by regenerated cellulose fibres could be achieved when 80% of the pre- and post-consumer denim wastes are collected. Therefore, the introduction of fibre recycling via regenerated cellulose fibres will also make a substantial impact on the cotton consumption for jeans production.

## 1. Introduction

Global textiles production for clothing and footwear is expected to grow to 102 million tons by 2030. At present the majority of textile products is disposed either by incineration or landfill. The need to reduce the negative impact on the environment and the substantial contribution of the textile production to global warming motivated the European Commission to formulate a strategy for a more sustainable and circular textile economy [1]. A central part of this strategy is the clear focus on recycling and reuse, which favours fibre-to-fibre recycling instead of downcycling. Thus technologies based on thermal or chemical processes are favoured, while mechanical opening and downgrading into non-woven products e.g., for thermal insulation of buildings is unwanted [2]. Functional multilayered clothing e.g., an outdoor jacket will require substantial redesign for recycling before such a study can be undertaken. Activities first will include reduction in number of different material used, selection of dyes and finishes to facilitate separation and fibre-to-fibre recycling [2].

A circular approach requires the involvement of all stakeholders along the full life cycle of a product and integrates aspects of design for recycling, sorting and separation [3] as well as localised approaches for more sustainable products [4]. In this work we present to the best of our knowledge for the first time a concept for circularity of cellulose-based textiles which integrates all relevant process steps from fibre production to textile regeneration. The high annual production of 24 million tonnes of cotton fibre requires 2.6% of the global water supply and represents one of the biggest consumers of pesticides and insecticides [5,6]. The field of indigo-dyed denim textiles was chosen as the representative case in this study because of the limited complexity with regard to fibre composition and dyes used. Additionally, we experimentally addressed the potential of reshaping indigo coloured fibres into coloured regenerated cellulose.

Circularity of cellulose-based textiles represents a particular challenge for two major reasons:-Cotton fibres are short staple fibres with a fibre length up to 50 mm. The direct fibre recycling would be favourable from the point of a life cycle analysis (LCA) [7]. However, the short fibre length and the high twist in the yarn prevent a simple re-opening of a cotton textile into individual cotton fibres with a final fibre length of 20 mm and higher. Shorter fibre lengths limit yarn production to high yarn counts with comparable low mechanical strength [8].-The complex chemical structure of cellulose prevents recycling via depolymerisation and polymerisation [9].

As a consequence, the most promising alternative to recycle the huge amounts of cellulose-based textiles will be their use for the production of regenerated cellulose fibres [10]. Different processes have been investigated for the production of regenerated cellulose fibres from textile wastes. In a comparative life cycle inventory (LCI) of different fibres and textiles, the regenerated cellulose fibres including viscose fibres could achieve a favourable position in the rating of greenhouse gas emissions, energy use in the raw material extraction and water use in the raw material extraction, even without consideration of textile wastes as cellulose feedstock [11]. The use of textile wastes for fibre production will lead to a further improvement of the LCA relevant profile of the regenerated cellulose fibres [12]. The use of ionic liquids as cellulose solvents has been proposed for the dissolution of cellulose fibre containing post-consumer wastes. Therein the ionic liquid 1-butyl-3-methylimidazolium acetate ([BMIM] [OAc]) or 1-allyl-3-methylimidazolium chloride (AmimCl) has been used as a solvent to separate cotton/polyester blends through dissolution of the cellulose component [13,14]. In case of cotton-based post-consumer wastes the process however requires a pre-hydrolysis of the cellulose cotton with sulphuric acid or cellulases to reduce the molecular weight as condition for complete fibre dissolution [15]. Aqueous solutions of LiOH/Urea and NaOH/urea or N-methyl-morpholine-N-oxide (NMMO) have been used for dissolution and regeneration of cellulose from post-consumer wastes [16,17]. In the recycling of post-consumer wastes, however the sensitivity of the cellulose-NMMO solutions for presence of traces of heavy metals e.g., Cu may cause problems due to initiation of exothermal reactions, which can result in an explosive reaction between cellulose and NMMO.

Cellulose recycling via regenerated cellulose fibres will also allow to recycle other cellulose based fibres e.g., flax and hemp as feedstock for regenerated fibre production [18]. These fibres can be grown in a moderate climate without a high demand of water for irrigation.

As we will demonstrate in this paper, the recycling of cellulose fibres via the route of regenerated cellulose fibres will increase the share of these fibres in textiles substantially. For this reason, regenerated cellulose fibres were chosen as fibre material in this study.

When a realistic approach for circularity of cellulosic pre- and post-consumer wastes is intended, the diversity of products, dyes, and finishes used to produce cellulose textiles must be considered [19].

Denim textiles were chosen as a representative case in this work because of its size in the market and the relative uniformity in material composition of the garment [20]. The global denim market has reached a volume of 90 billion dollars which corresponds to nearly 4 billion pieces [20]. 10% of the annual cotton production, respectively 2.5 million tons, goes into jeans production. The vast majority of these products is dyed with synthetic indigo, which holds an annual production of 70,000 tons [21]. Due to this unique situation, a number of approaches to recycle denim pre- and post-consumer wastes have been presented in the literature. A study to mechanically open denim textiles and spin the fibres to yarn presented very low yields of product and high losses in fibre material of 50–60% [8]. In addition, substantial shortening of the opened fibres to 10–20 mm fibre length was observed. The high losses and the fibre shortening make such an approach not feasible for a timely concept of circularity as after the second cycle almost 100% of the initial material will be gone into trash [22].

A more promising route to achieve efficient recycling of cotton cellulose goes the production of regenerated cellulose fibres. In this case a solution of the cellulose is shaped into new fibres [10]. Cellulose dissolution can be achieved using the viscose process by xanthogenate formation and dissolution in NaOH, by dissolution in NMMO or ionic liquids, or using NaOH/urea solution [16]. These processes are quite tolerant with regard to the presence of molecular disperse dyes e.g., vat dyes or reactive dyes, thus processing of coloured material is possible in principle [23,24]. The processing of indigo containing post-consumer wastes into blue coloured Ioncell fibres has been reported by Haslinger et al. [24]. However substantial loss of the dye and a change in colour tone were observed during the process, in addition significant dye leaching into the regeneration bath occurred.

The viscose fibre (CV) process was chosen in this study as a share of 79% of the global production of 7.1 million tons of regenerated cellulose fibres currently is manufactured via the xanthogenate process [25,26]. The production of mass coloured viscose fibres already has been reported in the literature. Through addition of finely powdered indigo dispersion coloured viscose fibres are obtained thus demonstrating the inertness of the oxidised indigo dye to the NaOH/CS_2_ solution [27,28].

Summarising, a considerable amount of research in the field of regenerated cellulose fibres is available in the literature; the studies however addressed a particular scientific question in detail. A scientific approach to consolidate the fragmented knowledge into a closed circular concept has not been investigated up to now. A relevant gap exists with regard to the knowledge about the behaviour of dyes during fibre-to-fibre recycling and the reuse of coloured material for production of new textile garment.

In this paper, we present to our knowledge for the first time a concept for circularity of indigo dyed fabric. Experimental and conceptual work will be presented to answer the following hypotheses and research questions:The viscose process is a robust technology. Information is required if and how indigo dyeing could be integrated into the production of regenerated cellulose fibres to serve as feedstock for denim products.During dope formation, the cellulose matrix dissolves and the indigo dye is released into the spin dope. Information is required if agglomeration of the molecular disperse indigo blue leads to coagulates which would prevent fibre spinning. Intensive bleeding of dye into the acidic coagulation bath also would prevent future technical implementation.For the denim market, dark blue fibres are needed. Regenerated fibres from denim textiles are light blue as dye loss may be expected during the use phase of garments. Could the colour depth of the regenerated fibres be increased at the stage of fibre production by addition of indigo dye to produce dark blue fibres, which then makes the environmentally unfavourable warp dyeing process obsolete?

The behaviour of indigo-containing viscose fibres in a simulated viscose process was studied experimentally. Quantitative analysis of the dye content in the fibre and regenerates, corresponding CIELab colour coordinates and colour strength in terms of Kubelka-Munk K/S values were determined and compared to conventional indigo dyeing on cotton. The direct addition of indigo to the spin dope was studied with the aim to produce a regenerate with sufficient indigo content for denim production without an additional dyeing step. Based on the experimental data model, calculations for the circular process are presented to provide an estimate for savings in terms of cellulose feedstock, chemicals, indigo dye, and water.

## 2. Experimental

### 2.1. Chemicals and Materials

Two sets of viscose fibres were used in the experiments, both donated by Kelheim Fibres GmbH, Germany, a fibre cable containing fibres of 1.7 dtex fineness as fibre cable (mass per length 150 ktex) (CV1 Danufil^®^ KTF), and staple fibres of fineness 1.7 dtex and length 40 mm (CV2 Danufil^®^ F). The fibre cable was used for the simulated indigo dyeing, while the staple fibre was used as model textile fibres from post-consumer wastes. The degree of polymerisation of the fibres was at 400 ± 53 glucose units per cellulose molecule [29]. The fibres were used without further purification, as preparations would be removed during indigo dyeing or alkaline steeping.

The dyestuff was obtained from DyStar Textilfarben GmbH & Co. KG, Frankfurt, Germany (“Dystar Indigo gran” and “DyStar Indigo Vat Solution” with analytically determined indigo content 23 wt%) and the de-aeration agent (alkylphosphate, Primasol^®^ NF) and dispersing agent (lignosulphonate, Setamol^®^ WS) were supplied by BASF AG (Ludwigshafen, Germany). Technical grade triethanolamine (85 wt%), sodium hydroxide (50 wt%), sulphuric acid (96 wt%) and sodium dithionite were purchased from Deuring GmbH & Co. KG (Hoerbranz, Austria). All other reagents were of analytical grade.

### 2.2. Preparation of Indigo Dyed Viscose

The dyeing was performed by the padding process on a ca. 10 m long tow of CV1 (mass of ca. 1 kg) at a nip pressure of 2 bar and roller speed of 1 m min^−1^. The tow was padded three times through 30 L dye liquor with a ca. 10 min pause between runs, and at the end, rinsed thoroughly with running soft water until the wash-off was colourless. Finally, the tow was allowed to dry in ambient atmosphere.

The dye liquor was formulated, as per standard recommendations, by mixing together 1.5 L of a “stock vat” formulation and 28.5 L of a “blank vat” formulation [30]. The stock vat consisted of 80 g L^−1^ indigo dye, 60 g L^−1^ sodium dithionite, 57.2 g L^−1^ NaOH, 4 g L^−1^ dispersing agent and 1 g L^−1^ de-aerating agent. The blank vat consisted of 2.21 g L^−1^ NaOH, 1.5 g L^−1^ sodium dithionite and 0.5 g L^−1^ dispersing agent. The two formulations were mixed immediately before the first padding operation and the pH (glass electrode) and redox potential (Pt electrode vs. Ag/AgCl/3M KCl reference) of the liquor were monitored at regular intervals. NaOH and sodium dithionite were added as needed, to maintain it at levels between 12.5 and 13 and below −740 mV respectively.

### 2.3. Cellulose Dissolution and Regeneration

Both the undyed CV2 and indigo-dyed CV1 were used in these experiments, after cutting them into short lengths of ca. 2 mm. The procedure employed was based on literature descriptions of a standard viscose process, and in general, consisted of the following sequence of steps [31,32]:

*Steeping:* About 3 g of cellulose was immersed in 47 g of 20 wt% NaOH at room temperature under regular stirring with a glass rod for 60 min, at the end of which about 31–32 g of the alkali was pressed out of the mixture.

*Xanthation:* Carbon disulphide (30 wt% on mass of cellulose) was added to the cellulose-NaOH mixture, and regularly stirred with a glass rod at room temperature for 60 min.

*Dissolution:* The xanthation mixture was transferred into 20 g of 8 wt% NaOH and stirred at room temperature for 1 h, and then stored overnight in a refrigerator at 8–12 °C.

*Regeneration:* The mixture from the previous step was taken out of the refrigerator and rested for a few hours for equilibration to ambient temperature, and then transferred into a regeneration bath containing 65 g of 10 wt% H_2_SO_4_ solution where it was stirred for 60 min.

*Neutralization/Washing:* The regenerated cellulose was filtered and rinsed with deionised water until the wash off exhibited a neutral pH as tested with pH paper, after which they were dried overnight in a laboratory oven at 60 °C.

In all experiments, the redox potentials of the reaction mixtures were measured immediately before refrigeration in the dissolution step and immediately before addition to the acid bath in the regeneration step.

Some variations to the above-described general procedure were employed in the following experimental sets, as described below:

#### 2.3.1. Continuous Monitoring of the Redox Potential

Separate measurements were performed with 6 g of undyed CV2 and indigo-dyed CV2. It was required to increase mixture volumes to allow for satisfactory immersion of the redox electrodes in the reaction mixes. Thus, the amount of NaOH in steeping step was increased to 144 g and there was no pressing out of alkali at the end of steeping. In the xanthation step, carbon disulphide was added in three stages in amounts of 0.9 g, 0.9 g and 1.2 g over 90 min. In the dissolution step, 100 mL of deionised water was added directly to the xanthation mixture to dilute the existing NaOH to a concentration of ca. 12 wt%. In the regeneration step, the dissolved cellulose was transferred into about 150 g of 33 wt% H_2_SO_4_.

#### 2.3.2. Variations of the Regeneration Bath

All experimental variables were as described for the general procedure except compositions of the regeneration bath, where the followings were employed:(a)10 wt% H_2_SO_4_(b)10 wt% H_2_SO_4_ + 10 wt% Na_2_SO_4_(c)10 wt% H_2_SO_4_ + 10 wt% Na_2_SO_4_ + 5 wt% ZnSO_4_(d)10 wt% H_2_SO_4_ + 10 wt% Na_2_SO_4_ + 5 wt% ZnSO_4_ + 2 wt% glucose

#### 2.3.3. Variations of Carbon Disulphide Amounts Added in the Xanthation Step

The indigo-dyed CV1 was the initial cellulose. The amount of carbon disulphide in the xanthation step varied between 45%, 60% and 75% on mass of cellulose.

#### 2.3.4. Regeneration from Mixture of Dyed and Undyed Cellulose Fibres

The initial amount of cellulose was made up of 25% by mass of dyed cellulose (i.e., indigo-dyed CV1) and 75% by mass of undyed cellulose (i.e., CV2). Further, two levels of carbon disulphide addition were employed, 30% and 75% on mass of cellulose.

#### 2.3.5. Addition of Pre-Reduced Indigo in the Viscose Process

The undyed CV2 fibres were used as the initial cellulose. Pre-reduced indigo was added after one hour into the dissolution step (in amounts equivalent to 1%, 3% and 5% on mass of cellulose) and the carbon disulphide amount in xanthation was 75% on mass of cellulose. Redox potentials of the reaction mixture were measured immediately before and after addition of pre-reduced indigo.

### 2.4. Analysis of Indigo Content

The indigo content in fibres and regenerates were determined by extraction of the dye from the specimens (e.g., 0.2 g) in 100 mL of an iron-triethanolamine complex at room temperature, and photometric evaluation of the dye extract at 407 nm in a 10 mm path length optical glass cuvette on a UV-VIS spectrophotometer (Model MCS 601, Carl Zeiss Spectroscopy GmbH, Jena, Germany). A calibration plot of absorbance vs. indigo dye in the complex (0–0.02 g L^−1^) yielded a coefficient of 0.706 ± 0.073, which was used in the determinations.

The iron-triethanolamine complex was formulated to contain 5 g L^−1^ FeSO_4_·7H_2_O, 10 g L^−1^ NaOH and 50 g L^−1^ triethanolamine. It was prepared by separately dissolving the iron salt and triethanolamine in small amounts of deionised water, adding the NaOH to triethanolamine, and mixing together the iron salt and triethanolamine-NaOH solutions before adding more deionised water to make up the final volume. The complex has a usable lifetime of ca. 60 min, so all extractions and measurements were finished within this time. The dye extraction from specimens required ca. 5–10 min after which the fibres were seen to become colourless.

### 2.5. Colour Measurement and Determination of K/S Value

The reflectance in the visible range (400–700 nm), with specular component excluded, was measured on a d/8 spectrophotometer with pulsed xenon lamps as light source (Model CM 3610d, Konica Minolta, Japan). The diameter of the measurement area was 8 mm. The reflectance data were transformed to L*, a* and b* coordinates of the CIE colour space (D65 illuminant, 10° observer) with the onboard software. The colour depth (K/S) was estimated with the Kubelka–Munk function (K/S = [(1 − R)^2^/2R]), where R is the fractional reflectance at the reflectance maximum (660 nm). These measurements were performed directly on the dyed fibres. However, in case of the regenerated particulates, the measurements were performed on specimens sandwiched between microscopy glass slides for fear of contaminating the device otherwise. The effect of the glass on results was determined by comparing measurements with and without the glass on fibres, and found to be negligible.

### 2.6. Determination of Indigo Particle Size

The crystallite sizes of the indigo, determined with powder X-ray diffraction, were used as an estimate of their particle size. The X-ray diffraction patterns (XRD) were recorded (at 25 °C) on a powder diffractometer (X’Pert PRO, Malvern PANalytical Inc., The Netherlands). The instrument was equipped with a *θ*/*θ* coupled goniometer in transmission geometry, programmable XYZ stage with well plate holder, and Cu-Kα_1,2_ radiation source with a focusing mirror. The incident beam side featured a 0.5° divergence slit, a 0.02° Soller slit collimator and a 1° anti-scattering slit; and the diffracted beam side featured a 2 mm anti-scattering slit, a 0.04° Soller slit collimator, a Ni-filter, and a solid-state PIXcel^1D^ detector. The patterns were recorded at a tube voltage of 40 kV, tube current of 40 mA, with a step-size of 0.013° 2*θ* in the angular range of 2°–40° 2*θ* at 400 s per step. The samples were milled and poured onto the sample holder to reduced preferred orientation.

The indigo crystallite sizes were determined from X-ray diffractograms with Scherrer’s Equation (1) [33,34]:τ = *K*λ(*β*cosθ)(1)
where τ is the thickness of the crystal perpendicular to the lattice plane represented by the XRD peak; *K*, a constant that depends on the crystal shape (0.89); λ, wavelength of the incident beam in the diffraction experiment (1.540598 Å); *β*, full width at half maximum (FWHM) in radians of the peaks corresponding to the (100), (10–2) and (210) lattice planes. Each of the peaks was fitted individually using the default settings in HighScore Plus v. 3.0e (Malvern PANalytical Inc., Almelo, The Netherlands). Whereas all three lattice planes were used in case of the dye alone, only the (100) and (10–2) were used for the dye in cellulose due to the overlap with a cellulose II reflection position at 2theta 26° and the indigo (210) lattice plane.

## 3. Results and Discussion

### 3.1. Indigo Dyeing of Regenerated Cellulose Fibres

In this study viscose fibres (CV) were chosen as feedstock for the recycling experiments as the vast majority of regenerated cellulose fibres are produced via this route. Despite the fact that hazards are related to the use of CS_2_ in the xanthogenation step, a favourable LCI profile of viscose fibres with regard to greenhouse gas emissions, water and energy consumption has been reported in the literature [11].

In case of denim recycling the warp of the fabric will contain indigo. Thus CV1 fibres were dyed with indigo according to the rope dyeing procedures used at present [30]. Indigo powder was reduced in a stock vat and a CV1 fibre cable was dyed similar to the technical process by one to three dips. One dip included immersion into the dyebath, squeezing off excess dyebath and air oxidation (greening). A large dyebath volume of 30 L was used to dye the CV1 cable (1 kg) in 1–3 dips (Figure 1). The initial pH in the dyebath was pH 13.07 and a redox potential of −765 mV was measured in the bath, which both dropped during the continuous dyeing to pH 12.08 and −746 mV at the end of the third dip. The redox potential was sufficient negative to maintain the indigo dye in its reduced form, despite the continuous oxidative load brought into the dyebath with the goods.

As expected, the samples showed a build-up in colour depth with increasing number of dips.

The colour of a 3 dip dyeing is shown in Figure 2a by means of a representative reflectance curve and the K/S value in the wavelength interval of 400 nm–700 nm. The increase in colour depth with number of dips is also shown in Figure 2b. The relatively large error bars for the average K/S values in Figure 2b are characteristic for the indigo dyeing process where short dyeing time and low dyebath temperature lead to minimum dye penetration and ring dyeing of the yarn, which leads to uneven dye distribution in the CV1 fibre cable.

The indigo content in the dyed viscose fibres was analysed with 30.11 ± 7.78 g kg^−1^ for three dips. The CIELab coordinates, the K/S value at 660 nm and the indigo content of the dyed viscose fibres are given in Table 1.

For the visual appearance of a dyeing, the CIELab coordinates are the decisive parameters. The K/S values and the CIELab coordinates given in Table 1 both demonstrate that the same shade of colour is obtained with indigo on CV1 fibres, when compared to cotton. The dyeing on viscose shows the expected CIELab coordinates for an indigo dyeing, thus replacement of cotton with indigo dyed viscose fibres will be a realistic approach. Differences in the relation between L* coordinate and K/S value can be explained with the different structure of the dyed material (fibres, yarn, fabric) and differences in the geometry of the devices used for reflectance measurement. In denim production, indigo dyeing is performed using a continuous pad-oxidation process which is repeated several times. Therefore, the processes are very sensitive to the conditions in the dyebath (pH, state of dye reduction, dye concentration, temperature, time of immersion) and on the fibre material. The CIELab coordinates can be used as indication for the stability of the dye under the chosen experimental conditions.

### 3.2. Compatibility of Indigo Dye with the Viscose Process

Indigo is a vat dye, which requires a redox potential lower than −700 mV (vs. Ag/AgCl, 3 M KCl reference) for reduction. Therefore, the reducing conditions present during the xanthation process and formation of the viscose dope could lead to dye reduction and agglomeration of the molecular disperse indigo dye and thus reduce the colour depth at a given dye concentration (Reaction scheme in Figure 3).

With the addition of CS_2_ to the alkalicellulose, xanthogenate and sulphidic compounds e.g., sulphide, dithiocarbonate are formed, thus the redox potential decreases (Figure 3) [37]. In Figure 3, the build-up of reducing conditions during the viscose process with undyed cellulose and indigo dyed cellulose (3dip) is shown.

In the aqueous NaOH (20 wt%), the initial redox potential near −100 mV decreases with addition of cellulose −500 mV which mainly is due to the presence of minor concentrations of reducing sugars and polysaccharides. The addition of CS_2_ leads to further reduction of the redox potential. With every addition of CS_2_ (step 2–4, Figure 3) the redox potential decreases further, however stabilises over −700 mV. Addition of water (step 5) leads to intake of dissolved oxygen and dilutes the reducing agents, thus the redox potential increases to −500 mV. Due to the excess of sulphides, a slow recovery then takes place and after 20 min a redox potential below −600 mV is reached again. The presence of indigo in the dyed sample does not lead to significant differences in the build-up of the reducing conditions. The minimum reduction potential for a successful indigo dye reduction is more negative than −700 mV, which was not reached in these experiments, consequently at the given experimental conditions no indigo dye reduction is expected to occur during the viscose process. In the experiments with undyed CV1 the formation of yellow sulphide-based products is observed which leads to the characteristic colour of the spin dope. In Figure 4, photographs of the experiments with undyed and indigo dyed CV1 are shown.

### 3.3. Dissolution and Regeneration of Indigo Dyed CV

For a technically feasible process concept, the presence of indigo in the cellulose feedstock must not interfere with the dissolution and regeneration process. No significant loss of indigo must occur during the viscose process, e.g., through degradation reactions. Contamination of the regeneration bath through bleeding of dye from the regenerated fibre into the acidic bath must be kept at a minimum. Thus, a series of dissolution–regeneration experiments was performed with dyed CV1 to investigate the compatibility of indigo with the viscose technology (Appendix A). A 3 dips indigo dyed viscose with a dye content of 30.11 ± 7.78 g kg^−1^ indigo was used as feedstock. After the step of cellulose regeneration, the CIELab coordinates, the K/S value and the indigo content of the regenerates were determined. The results are given in Table 2. The indigo content of the regenerates is presented in Figure 5.

The influence of the amount of CS_2_ used for preparation of the spin dope was investigated by use of different amounts of CS_2_ for the xanthogenation (experiments A–C in Table 2). Related to the mass of cellulose, an amount of 45 wt%, 60 wt% and 75 wt% CS_2_ was used in the experiments. The CIELab coordinates and indigo content of the regenerate from xanthogenation with 75% CS_2_ (expt. C in Table 2) are nearly identical to the dyed CV1 fibres used as feedstock. Any dye degradation, e.g., into the yellow isatin or dye agglomeration would lead to changes in colour coordinates and lower the indigo content in the regenerates. The results demonstrate the stability of the molecular disperse indigo under the experimental conditions of a viscose process, which is an essential condition to regenerated indigo containing denim textiles via the route of CV production. Remarkably, the regenerates which had been obtained from xanthogenates using 45 wt% CS_2_ or 60 wt% CS_2_ showed darker colour and higher indigo content. Due to the colour variation in the dyed viscose cable, the differences between the regenerates and the feedstock material are not significant; however, the increase in dye content between the regenerate from 75 wt% CS_2_ to 60 wt% or 45 wt% is significant. This could be due to losses of cellulose during the simulated viscose process; however, large-scale experiments will be needed to confirm this assumption.

Besides the stability of the molecular disperse dye, also complete and homogenous cellulose dissolution must be achieved in the viscose spin dope to avoid uncontrolled and uneven distribution of the dye in the regenerate. Mixtures of dyed CV1 fibres (25 wt%) and undyed fibres CV2 (75 wt%) were tested in dissolution–regeneration experiments followed by colour measurement and indigo analysis (experiment D and E in Table 2). The regenerates were homogenously coloured products and their colour depth in terms of K/S values 12.31 ± 0.28 (experiment D, Figure 5) 9.08 ± 0.41 (experiment E) and the indigo content of 10.78 ± 1.35 g kg^−1^ and 11.91 ± 0.53 g kg^−1^ respectively were at the expected levels.

Colour changes also could be induced during the cellulose regeneration step in the acidic coagulation bath. Thus, a series of experiments with use of different cellulose regeneration baths was performed. Besides regeneration in 10 wt% H_2_SO_4_ (experiment F, Table 2), the influence of Na_2_SO_4_, ZnSO_4_ and glucose (experiments G–I, Table 2) on colour shade and indigo content were studied experimentally. Na_2_SO_4_ is formed in the regeneration baths as follow-up product of the neutralisation of NaOH from the spin dope. ZnSO_4_ is found in regeneration baths where retarded cellulose regeneration is intended to produce modal type full-mantled fibres and glucose was added to evaluate the influence of reducing oligosaccharides on the colour of the dye containing regenerate. The comparison of the CIELab coordinates, the colour strength in terms of K/S and of the indigo content of the regenerates demonstrated the compatibility of the indigo containing spin dope with regeneration baths of different composition, which is a condition for the technical robustness of the process.

Despite the small scale of the experiments more than 85.3 ± 3.0% of the fibre mass could be recovered as regenerate in the experiments F–I.

### 3.4. Addition of Indigo to the Spin Dope

When post-consumer waste is used as a feedstock, the indigo content of the raw material is much lower compared to the dark dyeings which are used for denim production. To eliminate the later step of warp yarn dyeing completely, the dye addition has to be integrated into the stage of fibre production. Addition of indigo into the fibre dope can be made in form of finely dispersed oxidised indigo which then follows the procedure of a standard dyeing with dispersed pigment dyes [28]. The development of the characteristic indigo shade and the colour depth require presence of molecular disperse dye instead of a dye pigment. Thus, in this work the direct addition of an alkaline solution of leucoindigo (hydrogenated indigo) with an indigo concentration of 23 wt% indigo was investigated. The redox potential measurements in Section 3.2 indicated that the reduction potential in the spin dope which will be insufficient to reduce indigo, however will permit addition of leucoindigo to the spin dope without oxidation to indigo. In these experiments undyed CV2 served as feedstock and three different concentrations of indigo were realised in the spin dope by addition of leucoindigo solution (Appendix A). The detailed conditions are given in Table 3. For comparison of colour shade and K/S value also data from technically dyed denim (samples N1–N3) and from indigo pigment dyed viscose fibres (samples M1 and M2) are displayed.

With increasing amount of added indigo, the colour depth of the regenerate and the indigo content in the regenerates increase.

For an assessment of the CIELab coordinates it is important to consider that a number of factors influence the outcome of the final shade. This includes the structure of the material e.g., powder, fibre, yarn, fabric, the reducing agent used to prepare the leucoindigo as well as the dyeing conditions e.g., slasher or rope dyeing, concentration of indigo in the dyebath, dyebath pH, state of reduction and presence of dispersed indigo in the dyebath. Thus, the colour coordinates in Table 2 demonstrate the chemical stability of the indigo molecule during the alkaline or acidic conditions of the fibre spinning process. A detailed adjustment of the colour to a certain shade must be undertaken during technical optimisation.

The colours of the samples J–L are near to the shade of reference dyeings on cotton (N1-3), with a slightly less negative b* coordinate. Besides reference dyeings from conventional indigo dyeing, also colour data for a spun dyed regenerated cellulose fibre are displayed which had been mass coloured by addition of dispersed indigo pigments (sample M1 and M2). The CIELab coordinates of samples J–L are very near to these dyeings as well as to the other reference dyeings given in Table 3. However, for dyeing J, only 13 g kg^−1^ indigo were required to obtain a similar colour depth to the samples M1 and M2 for which a dye content of approximately 3% was given.

No bleeding of blue dye into the regeneration bath was observed which is of importance for large-scale production. In the Appendix A, photographs of the regeneration baths are given (Appendix A). Release of significant amounts of indigo into the regeneration bath would lead to unproductive dye loss and would require additional treatment to remove the dispersed indigo before Na_2_SO_4_ is collected from the regeneration bath.

### 3.5. Determination of Indigo Crystallite Size

Vat dyes exhibit a characteristic dependency of the colour shade and the intensity on their particle size. Therefore, the direct use of a finely milled vat dye pigment as colorant yields dyeings with comparable low colour yield. By reduction, dissolution of the dye particle is achieved and the molecular disperse dye then forms during reoxidation in the cellulose matrix. This leads to smaller particle size and development of the full colour depth and the typical indigo colour. The determination of the indigo crystallite size was performed to obtain an indication for possible recrystallisation and larger particle formation during xanthogenation and cellulose regeneration. Thus, crystallite sizes of indigo in dyed CV and of solid indigo dye were estimated from the respective X-ray diffractograms (Figure 6) (Appendix A).

The indigo polymorph B was detected in all samples: Key reflections are (100), (10–2) and (210) lattice planes [38,39]. Only (100) and (10–2) were used for the crystallite size estimation of indigo in the samples due to the overlap with cellulose II reflection positions at 2theta 26° and the indigo (210) peak position.

In synthetic indigo powder the average crystallite size has been calculated with 450 Å. A reduction in crystallite size occurs as a consequence of the reduction and reoxidation step during indigo dyeing. Thus, in the dyed CV2 (CV dyed in Figure 6) a lower crystallite size of 270 Å is observed. The dissolution and regeneration of the cellulose in the viscose process do not lead to a significant change in the size of the indigo crystallites. The smallest average crystallite size is indicated for regenerates, which had been obtained by direct addition of reduced indigo to the spin dope (samples J–L in Figure 6, Table 3). The results demonstrate that no recrystallisation of indigo to larger dimensions had occurred. Uncontrolled growth of the dye crystals could lead to blocking of the spinneret as well as to unwanted changes in colour shade and colour depth during the step of fibre formation. During filtration of the spin dope larger aggregates will be removed before fibre spinning; however, the amount of filtered material e.g., indigo particles must be kept at a minimum to prevent rapid filter blocking.

## 4. Process Balances—Model Calculations

### 4.1. Impact of Circularity in Denim on Cotton Consumption

With implementation of circularity for cellulose textiles via the path of regenerated cellulose fibres a major part of cotton consumption will be substituted by regenerated cellulose fibres. A scheme of the concept discussed in this work is presented in Figure 7. Indigo containing pre-consumer and post-consumer wastes will be collected and used for production of indigo-coloured regenerated cellulose fibres, which then are used both as warp and weft in denim production. In case of denim recycling, the indigo dye will be embedded into the cellulose structure of the dyed cotton and thus will also be present in the regenerate.

The recycling of cellulose based textiles into regenerated cellulose fibres will lead to a substantial impact on the raw material production. In case of cellulose recycling via regenerated cellulose fibres, the impact on the cotton consumption will depend on the proportion of collected material and on the efficiency of the fibre regeneration process. From the data given in this paper, an estimation of the cellulose recycling through formation of regenerated cellulose fibre can be made as a function of the share of material collected as pre-consumer and post-consumer wastes. Denim textiles can be identified and collected from both fractions rather easily thus a high percentage of these wastes would be available for recycling.

A flow scheme of the two most important steps of a circular cellulose fibre process is shown in Figure 8a. The reduction in use of fresh cotton fibre material *m_Co_* (kg) to produce 100 kg of material will depend on the proportion of material that will be collected for regeneration and on the efficiency of the fibre regeneration process. Based on the share of collected denim wastes *η_coll_* (%) and the efficiency of the viscose process *η_CV_* (%), the amount of regenerated cellulose fibres *m_CV_* (kg) and of cotton *m_Co_* (kg) for 100 kg fibre production in the circular concept can be calculated according to Equations (2) and (3).
(2)$$m_{CV}=100*\eta_{CV}*\eta_{coll}*10^{-4}$$
(3)$$m_{Co}=100-m_{CV}$$

In Figure 8b, the dependency of the required amounts of cotton and regenerated cellulose fibres on the proportion of collected denim waste and the efficiency of the viscose fibre process is given.

When 60 wt% of the material used for denim stems from recycled material, the share of reused CV fibres reaches 50 wt% of the total cellulose fibre consumption. This corresponds to a reduction in cotton consumption for denim production by 50 wt%, the accurate figure being dependent on the efficiency of the viscose process. When 80 kg of wastes are collected (*η_coll_* = 80%) for CV production and the efficiency of the viscose process *η_CV_* is at 90%, the cotton consumption will decline to 28% and the CV fibres will take a share of 72% of the material used for denim production. In case of denim products this reduction would be equivalent to more than 1.5 million tons of cotton per year.

For comparison, also the losses of cotton in a purely mechanical recycling concept during opening and spinning are given in Figure 8b [40]. Even when the amounts of collected material reach the theoretical maximum of 100%, a replenishment of 50% will be required for each cycle.

### 4.2. Combination of Indigo Dyeing and Viscose Fibre Technology

Based on the experimental results presented above, a general scheme for denim recycling and integration of the indigo dyeing technology into the viscose fibre process can be drawn (Figure 9). Representative cases for the indigo concentration in the different streams of material are given.

In Figure 9 a dark dyeing of the warp yarn with 30 g kg^−1^ indigo has been chosen as an example. During garment wash of denim and consumer use, the dye content of the warp reduces to e.g., 5 g kg^−1^ indigo. In a typical denim fabric, the undyed weft yarns comprise approximately 34 wt% of the total mass, while the dyed warp yarns make up 66% of the fabric weight [41]. Fabric material collected from the post-consumer waste stream thus will consist of undyed weft yarn (approx. 34 wt%, 0 g kg^−1^ indigo) and dyed warp yarn (approx. 66 wt%, 5 g kg^−1^ indigo).

Based on the efficiency both of the wastes collection *η_coll_* and of the fibre regeneration *η_CV_*, fresh cellulose has to be added to the process (cotton consumption in Figure 8b). In case of *η_coll_* = 80% and *η_CV_* = 90%, 31 kg of new fibre material and 80 kg of denim waste will be processed to obtain 100 kg of regenerate.

The undyed fresh fibres and part of the collected wastes will be processed to regenerate with low dye content (stream A) which will be used for the weft yarn. The dye content in these fibres then will be at 2.5 g kg^−1^.

The compatibility of the viscose process for the direct addition of an alkaline leucoindigo solution has been demonstrated in this work. The production of dark shades of indigo dyed viscose fibres forms a second stream fibre material (stream B), which leads to production of fibres for the warp yarn. In the representative example, 49 kg of recycled cellulose material is processed to deliver 44 kg of dyed viscose fibres. By addition of reduced indigo to the spin dope, the dye content in these fibres is adjusted to 30 g kg^−1^ indigo (similar to sample K in Table 3).

A characteristic property of the warp yarn in denim is the so-called “ring dyeing”. The core of the dyed yarn must remain undyed, which is a condition to achieve the desired faded look during the garment wash. The ring dyeing effect in the proposed concept will be achieved mechanically at the stage of spinning by production of a core-shell yarn with the dark fibres (stream B) forming the shell and light fibres (stream A) forming the core of the yarn. Such techniques are already widely used, e.g., to cover elastomer yarns with other fibres [42]. By such an approach the later dyeing of the warp yarn can be missed out. This will lead to substantial savings of chemicals and water.

### 4.3. Savings through Omission of Yarn Dyeing

The combination of fibre recycling and dyeing makes the later dyeing of denim yarn unnecessary. The step of yarn colouration contributes substantially to the carbon and water footprint of jeans [43]. The chemical and water consumption of a representative indigo yarn dyeing plant has been summarised by Abdelileh et al. [36]. For a daily production of 15,000 kg of dyed warp yarn, a consumption of 75,000 L of water and 50–126 kg dithionite and approximately the same amount of NaOH have been estimated [44].

The potential savings of a process which omits this step can be illustrated by the calculations given in Table 4 where the water and chemical consumption for a technical plant per day and within a year (250 working days), and for an annual production of 1.5 million tons of dyed warp yarn are shown [21,36].

The estimates indicate that replacement of the indigo warp dyeing process by spun dyed regenerated viscose directly leads to substantial savings in water and chemicals.

Minor savings in indigo consumption also could be expected considering the indigo dye brought into the system with the recycled material. This amount of indigo however depends on the intensity of the previous garment wash and bleach, as well as on the consumers use. As a rough indication, the reuse of material with an indigo content of 2.5 g kg^−1^ indigo to produce dyed material with a content of 30 g kg^−1^ indigo would lead to a nearly 10% reduction in indigo consumption

## 5. Conclusions

Circularity of textile products will be successful only when fibre material, dyes and additives are considered in a coordinated approach. Because of their unique appearance, denim products can be collected and separated comparably easily which makes them an ideal candidate to establish such a model case for circularity of cellulose textiles.

The experimental results to assess the behaviour of indigo in the viscose fibre process demonstrated that indigo exhibits sufficient stability under the conditions of alkaline steeping and xanthogenation. The finely divided dye retains its position in the fibre matrix during cellulose regeneration in sulphuric acid and no bleeding of dye into the regeneration baths was observed.

The indigo content in post-consumer wastes is lower than required for warp yarn in denim. Thus, direct addition of reduced indigo to the spin dope was investigated. Dark blue fibres were obtained through addition of reduced indigo solution to the spin dope. Colour depth and shade of the dyeings were within the required colour range for indigo dyed cellulose yarns. When such fibres are spun into core/shell yarns, ring dyed yarns can be obtained without use of the polluting yarn dyeing process. The proposed concept thus results in substantial savings in cotton consumption, and also reduces the environmental impact of the denim production by omitting the step of yarn dyeing. Scale up of the technique into the process of viscose fibre production is part of the ongoing research activities.

Future work should address several topics:Extension of the concept to other techniques of regenerated cellulose fibre production e.g., use of ionic liquids.Investigation of the behaviour of sulphur dyes during fibre regeneration as these dyes often are used for topping or bottoming of indigo dyed yarn.Evaluation of techniques to separate other minor fibre components e.g., elastane fibres, polyester sewing threads during fibre regeneration. A technique to separate elastane from other fibres already is in experimental evaluation.

## Figures and Tables

**Figure 1 polymers-14-05280-f001:**
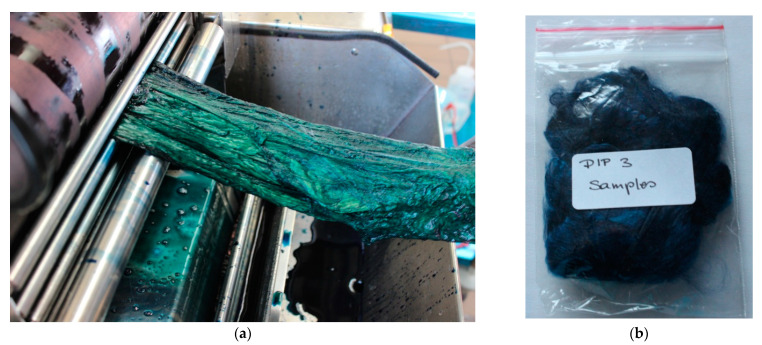
Dyeing of CV1 fibres (**a**) fibre cable after immersion into the dyebath, before squeezing off surplus of dyebath, (**b**) 3 dip indigo dyed sample.

**Figure 2 polymers-14-05280-f002:**
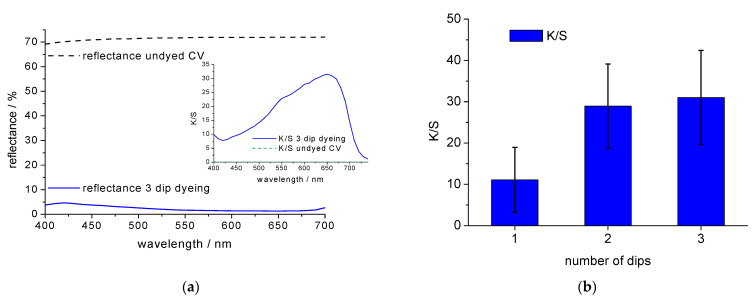
Representative examples for (**a**) reflectance curve and K/S value between 400 and 700 nm (inserted picture) for a 3 dip indigo dyed CV1 and for undyed CV1, and (**b**) K/S of dyed CV1 as function number of dips.

**Figure 3 polymers-14-05280-f003:**
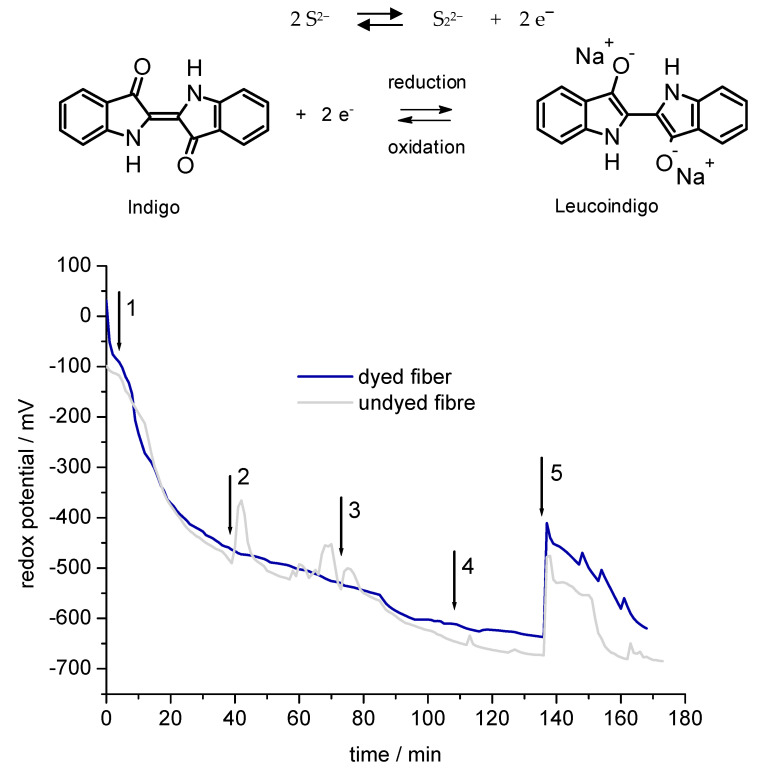
Reaction scheme of potential indigo reduction during xanthation and redox potential in viscose process as function of time, (1) addition of cellulose to NaOH, (2) addition of CS_2_, (3) addition of CS_2_, (4) addition of CS_2_, (5) addition of water; (

) undyed CV1, (

) 3 dip dyed CV1.

**Figure 4 polymers-14-05280-f004:**
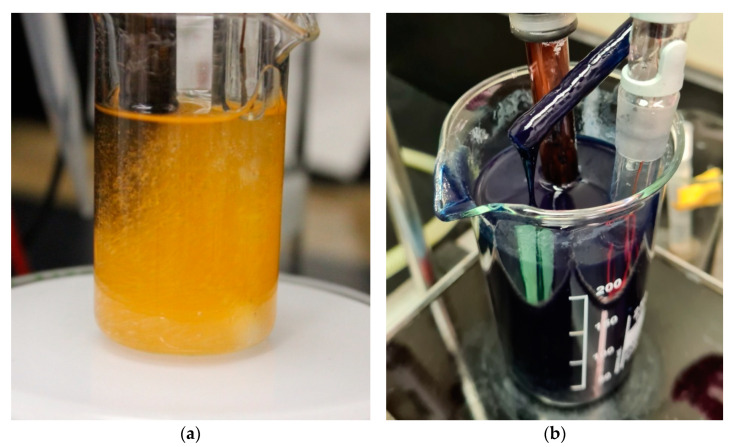
Photographs of cellulose dissolution experiments with measurement of redox potential during the viscose process (**a**) undyed CV2, (**b**) indigo dyed CV1.

**Figure 5 polymers-14-05280-f005:**
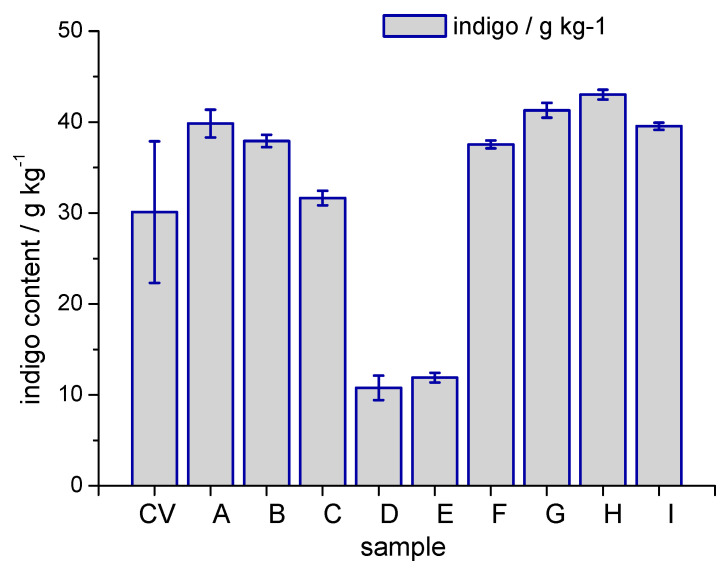
Indigo content in regenerated cellulose samples, (CV) indigo dyed CV1, (A) 45 wt% CS_2_, (B) 60 wt% CS_2_, (C) 75 wt% CS_2_ in xanthogenation, (D) mixture of 75% undyed CV fibres with 25% dyed fibres, 30 wt% CS_2_, (E) mixture of 75% undyed CV fibres with 25% dyed fibres, 75 wt% CS_2_, (F–I) use of different composition of regeneration baths (for details see Table 2).

**Figure 6 polymers-14-05280-f006:**
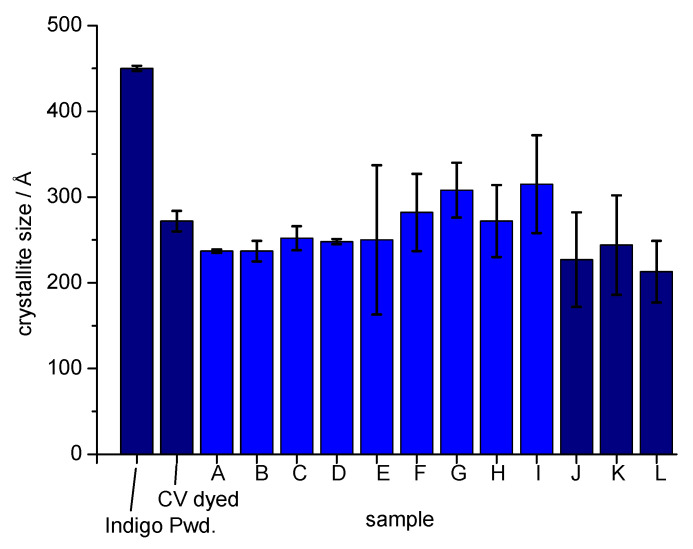
Crystallite size of indigo powder (Indigo Pwd.), dyed viscose CV2 (CV dyed) and indigo containing regenerates A–L (Table 2 and Table 3). Samples J–L were dyed with addition of leucoindigo solution.

**Figure 7 polymers-14-05280-f007:**
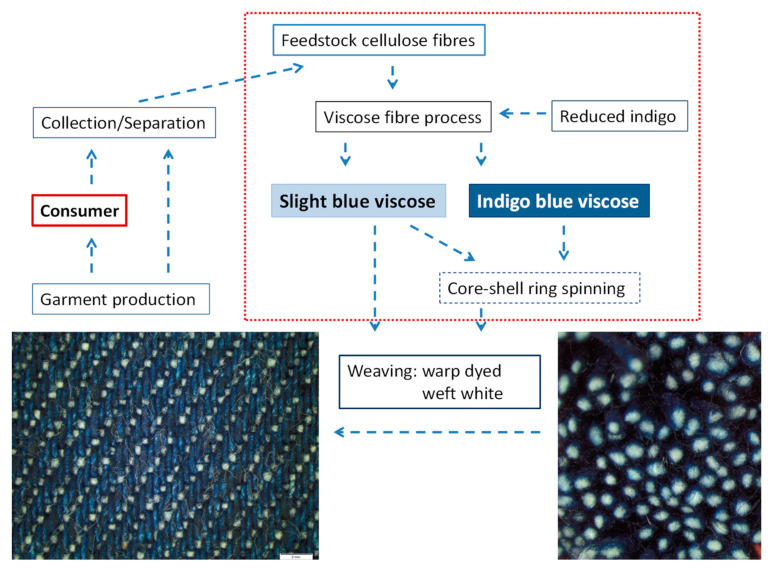
Schematic presentation of a concept for circularity in pre-consumer and post-consumer denim textiles through production of spun-dyed denim.

**Figure 8 polymers-14-05280-f008:**
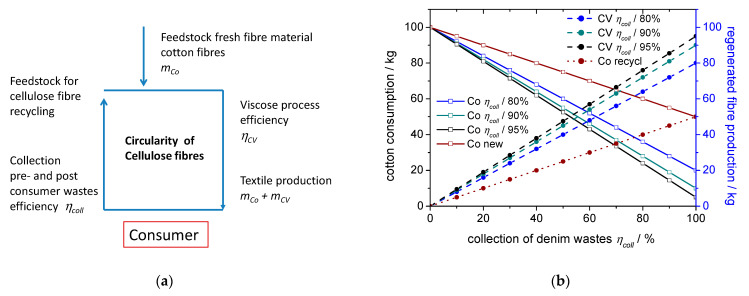
(**a**) Basic scheme for material flow in a circular concept based on regenerated cellulose fibres; (**b**) amount of cotton and regenerated CV for production of 100 kg yarn as function of proportion of collected denim wastes *η_coll_* and the efficiency of the viscose process *η_CV_*.

**Figure 9 polymers-14-05280-f009:**
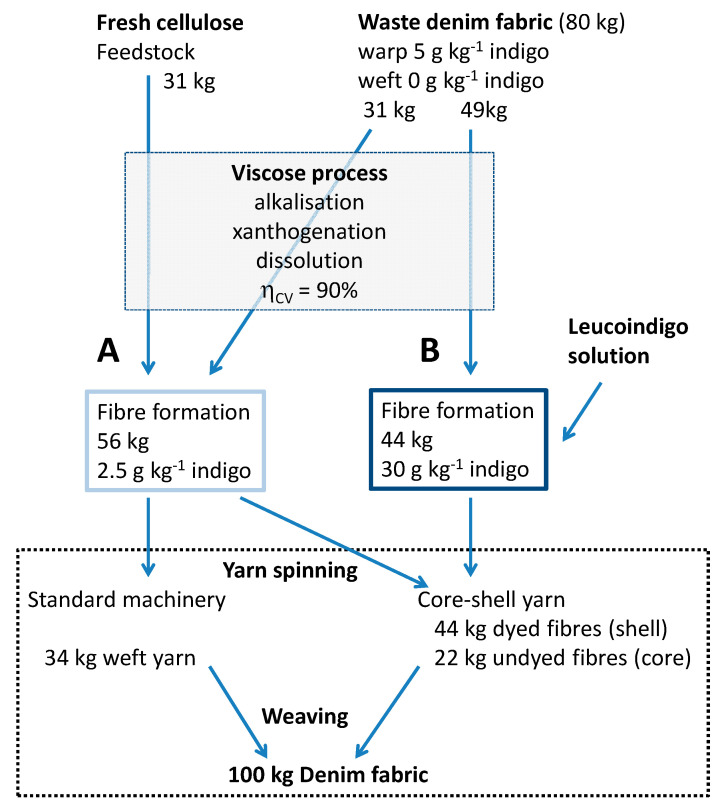
Flow scheme for recycling of indigo containing cellulose fibres to produce 100 kg of denim fabric. Relevant stages presented from fibre dissolution to yarn spinning.

**Table 1 polymers-14-05280-t001:** CIELab coordinates and K/S value at 660 nm of indigo dyed CV1 fibres and reference dyeings on cotton (Co) yarn or fabric.

Sample	CIELab Coordinates			K/S
	L*	a*	b*	
CV1 1 dip	38.41 ± 16.27	−2.70 ± 1.69	−19.41 ± 7.04	11.09 ± 7.85
CV1 2 dips	21.47 ± 7.52	0.19 ± 0.70	−19.82 ± 7.00	28.93 ± 10.22
CV1 3 dips	18.10 ± 7.09	0.52 ± 1.00	−15.08 ± 5.87	31.02 ± 11.39
Co ^a^	41.82 ± 1.28	−1.89 ± 0.21	−22.08 ± 0.62	4.15 ± 0.21
Co ^a^	30.56 ± 0.38	0.40 ± 0.14	−21.62 ± 0.04	8.1 ± 0.00
Co ^b^	20.00 ± 0.20	1.23 ± 1.80	−13.71 ± 0.23	
Co (Indigo pwd.) ^c^	39.1	−3.7	−24.6	11.01
(Indigo sol.) ^c^	29.6	−1.4	−23.1	18.87
Undyed CV1	87.84 ± 0.69	−0.12 ± 0.07	0.74 ± 0.29	0.06 ± 0.01

^a^ Fabric, 1% and 3% indigo pwd., RT, exhaust dyeing [35], ^b^ rope dyeing, 1.6 g L^−1^ indigo, 4 dips pad-oxidation (Avondale, AL, USA), ^c^ fabric, 3 dips pad-oxidation dyeing, 3 g L^−1^ indigo pwd, 6 g L^−1^ indigo sol. [36].

**Table 2 polymers-14-05280-t002:** CIELab colour coordinates, K/S and indigo content of samples prepared from dyed CV fibres by different dissolution and regeneration experiments.

Experiment	Variation	CIELab Coordinates			K/S	Indigo Content
	wt%	L*	a*	b*		g kg^−1^
A	45 CS_2_	14.54 ± 0.10	0.51 ± 0.09	−9.31 ± 0.21	34.22 ± 0.25 ^a^	39.84 ± 1.53
B	60 CS_2_	14.43 ± 0.30	0.94 ± 0.11	−8.31 ± 0.16	31.94 ± 1.30 ^a^	37.92 ± 0.68
C	75 CS_2_	18.71 ± 0.42	0.66 ± 0.07	−14.13 ± 0.04	27.24 ± 1.19 ^a^	31.65 ± 0.80
D	25 dyed CV1, 75 undyed CV2, 30 CS_2_	29.91 ± 0.25	−1.65 ± 0.03	−15.97 ± 0.08	12.31 ± 0.28 ^b^	10.78 ± 1.35
E	25 dyed CV1, 75 undyed CV2, 75 CS_2_	34.31 ± 0.64	−2.59 ± 0.09	−16.06 ± 0.25	9.08 ± 0.41 ^b^	11.91 ± 0.53
F	10 H_2_SO_4_	15.25 ± 0.33	0.31 ± 0.13	−9.41 ± 0.25	33.05 ± 1.25 ^a^	37.54 ± 0.43
G	10 H_2_SO_4_, 10 Na_2_SO_4_	14.54 ± 0.67	0.39 ± 0.20	−9.15 ± 0.33	35.02 ± 1.56 ^a^	41.30 ± 0.82
H	10 H_2_SO_4_, 10 Na_2_SO_4_, 5 ZnSO_4_	16.32 ± 1.67	0.31 ± 0.19	−9.87 ± 0.93	29.83 ± 4.08 ^a^	43.02 ± 0.54
I	10 H_2_SO_4_, 10 Na_2_SO_4_, 5 ZnSO_4_ 2 glucose	14.36 ± 1.01	0.39 ± 0.15	−8.27 ± 0.31	34.49 ± 3.66 ^a^	39.54 ± 0.40
Feedstock	CV1 3 dips	18.10 ± 7.09	0.52 ± 1.00	−15.08 ± 5.87	31.02 ± 11.39 ^b^	30.11 ± 7.78

^a^ 640 nm, ^b^ 660 nm.

**Table 3 polymers-14-05280-t003:** Indigo colour depth in wt% indigo related to cellulose, CIELab colour coordinates, K/S and indigo content of samples J–L prepared by addition of leucoindigo solution to the viscose spin dope, M1 and M2 spun dyed viscose with indigo pigment, N1–N3 conventional indigo dyeings on cotton as reference experiments were used as feedstock for samples J–L.

Experiment	Indigo Added	CIELab Coordinates			K/S	Indigo Content
	%	L*	a*	b*		g kg^−1^
J	0.81 ± 0.05	18.22 ± 0.41	−1.18 ± 0.32	−8.66 ± 0.81	28.08 ± 1.52	13.01 ± 0.89
K	2.28 ± 0.16	14.30 ± 0.84	−0.38 ± 0.28	−5.77 ± 0.45	32.47 ± 4.33	29.61 ± 2.04
L	4.29 ± 0.16	13.76 ± 1.49	0.76 ± 0.38	−4.98 ± 1.80	28.67 ± 4.11	50.12 ± 2.18
Reference dyeings						
M1	2.87	19.38 ± 0.27	3.21 ± 0.42	−17.80 ± 0.82	23.52 ± 0.36	
M2	3.27	17.86 ± 0.36	2.62 ± 0.05	−15.69 ± 0.45	24.14 ± 0.46	
N1 ^c^	-	20.00 ± 0.20	1.23 ± 1.80	−13.71 ± 0.23		
N2 ^d^	(indigo pwd.)	39.1	−3.7	−24.6	11.01	
N3 ^d^	(indigo sol.)	29.6	−1.4	−23.1	18.87	

^c^ rope dyeing, 1.6 g L^−1^ indigo, 4 dips pad-oxidation (Avondale, AL, USA), ^d^ fabric, dyeing with 3 dips pad-oxidation, RT, 3 g L^−1^ indigo pwd, 6 g L^−1^ indigo sol [36].

**Table 4 polymers-14-05280-t004:** Representative case for the production capacity of a yarn dyeing range and corresponding consumption of water, dithionite Na_2_S_2_O_4_ and NaOH in indigo dyeing per day and year, and extrapolations for the global denim production.

		Technical Dyeing Range		Global Production
Period		1 Day	Year	Year
Production	tons	15	3750	1,500,000
Water	m^3^	75	18,750	7,500,000
Na_2_S_2_O_4_	tons	0.050–0.126	12.5–31.5	5000–12,600
NaOH	tons	0.050–0.126	12.5–31.5	5000–12,600

## Data Availability

The data presented in this study are available on request from the corresponding author.

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
