# Peer review of "Dope Dyeing of Regenerated Cellulose Fibres with Leucoindigo as Base for Circularity of Denim"

_polymers, 2022, doi:10.3390/polym14235280_

Round 1

Reviewer 1 Report

This work presents "the position of regenerated cellulose fibres in the circularity of  

indigo dyed jeans", which is interesting and suitable for the publication in Polymers journal after addressing the following points.

1.      Why did the authors select two sets of regenerated viscose fibres (CV1 and CV2)? This seems unjustified as the author stated that A more promising route to achieve efficient recycling of cotton cellulose goes the production of regenerated cellulose fibres, please see the introduction part - lines 92-93.

2.      Add some references to the introduction part - lines 104-106.

3.      Full specification of the reference indigo dyed cotton samples (CO or N1-N3) should be included?

4.      The description of the conventionally indigo dyeings on cotton should be justified as reference experiments. How much indigo dye % was and which dyeing method used? The K/S values and the CIELab coordinates of CV fibres, listed in Tables 1 and 3 are questionable, when compared to CO yarn or fibers.

5.      Table 4. Representative case for the production capacity of a yarn dyeing range and corresponding consumption of water, reducing agent and caustic soda in indigo dyeing per day and year, and ex-trapolations for the global denim production should be referenced.

Reviewer 2 Report

Dear authors,

The data provided here for the article entitled "The position of regenerated cellulose fibres in the circularity of indigo dyed jeans" are interesting; however, I am offering some comments throughout the manuscript.

(1) The title of the manuscript does not comply with the present study. It should be revised.

(2) Abstract is more related to the object and methodology, but not to the results. The readers would like to grab all of your key findings after having a look at the abstract. But the recent abstract looks like a combination of some statements. The abstract must be improved.

(3) The introduction part is unprofessional; there is a lack of consistency between lines and paragraphs, and it really needs to be revised very carefully.

(4) The main gap of the work is totally absent in the Introduction. The last paragraph of the Introduction should provide information (only) about the science gap in the previous studies and what motivates you to do this review with the objective of the study.

(5) The authors have used “fiber” and “fibre”. It is recommended to maintain consistency throughout the manuscript.

(6) The authors have used “color” and “colour”. It is recommended to maintain consistency throughout the manuscript.

(7) All supplementary Figures are not cited in the main manuscript.

(8) All short forms are not abbreviated. It is recommended to use abbreviation first and then continue in a short form.

(9) The conclusion must be re-written with conclusive findings and by retaining coherence.

(10) References should be according to the journal template.

(11) I am not an English speaker, but I found many typos and grammatical errors throughout the manuscript. These must be corrected and revised.

Reviewer 3 Report

The Thomas Bechtold et al. manuscript deals with the recycling of cellulosic materials. When studying the material of the manuscript, the question arises - what is the novelty of the work and its scientific significance? at the moment, fabrics of this kind are easily processed into non-woven materials, the demand for which is constantly growing. What are the advantages of the described method compared to obtaining nonwoven materials? It is not entirely clear how to recycle products containing an insulated inner layer, and the number of such products in the Nordic countries is significant? The theoretical part, in my opinion, can be improved and expanded. The methodological part is presented at a good level. The main part contains a large number of blots, the quality of the drawings needs to be improved.   In the Introduction, the authors raise the issue of obtaining, using and processing cellulose fibers and products from them. The volumes of cultivation and consumption of natural fibers - cotton are considered. It is also mentioned that cotton requires large amounts of water. unfortunately, the authors overlook such popular natural fibers as flax and hemp and the possibility of repeated processing of such fibers. Authors should pay attention to works of this type, for example - https://doi.org/10.3390/fib10050045 Line 42. Length of cotton fibers up to 5 cm. Line 73. "Cu may cause problems due to initiation of exothermal reactions." - an unprepared reader may not immediately understand this expression. Line 81. "A share of nearly 10% of the annual cotton production of 25 million tons goes into jeans production." - an unfortunate expression, it is better to rephrase (2.5 million ....). 2.1. Chemicals and Materials. You must specify the degree of polymerization for the samples. Was fiber sizing removed and how? Line 272-274. Delete! Figure 2. You need to increase the font size in the figure. Lines 437-444. The samples under consideration have a different structure. Lines 841-848. When filtering large crystals must be excluded from the spinning dope?!

Round 2

Reviewer 3 Report

The authors took into account all the comments and made the appropriate changes to the manuscript. Further, the manuscript may be reviewed by the editor for acceptance for publication.

Lines 184, 185. "± 53 glucose units per cellulose molecule" - I recommend deleting it.